



# Origin and role of non-skeletal carbonate in coralligenous build-ups: new geobiological perspectives in the biomineralization processes

Mara Cipriani[1], Carmine Apollaro[1], Daniela Basso[2,3], Pietro Bazzicalupo[2], Marco Bertolino[4], Valentina Alice Bracchi[2,3], Fabio Bruno[5], Gabriele Costa[6], Rocco Dominici[1], Alessandro Gallo[5], Maurizio Muzzupappa[5], Antonietta Rosso[3,7], Rossana Sanfilippo[3,7], Francesco Sciuto[3,7], Giovanni Vespasiano[1], Adriano Guido[1*]

[1] Department of Biology, Ecology and Earth Sciences, University of Calabria, Via P. Bucci, cubo 15b, 87036, Rende, Cosenza, Italy;
[2] Department of Earth and Environmental Sciences, University of Milano-Bicocca, Piazza dell'Ateneo Nuovo, 1, 20126, Milan, Italy;
[3] CoNISMa-Italian National Inter-University Consortium for Sea Sciences, Piazzale Flaminio, 9, 00196, Rome, Italy;
[4] Department of Earth, Environmental and Life Sciences, University of Genoa, 16126, Genoa, Italy;
[5] Department of Mechanical, Energy and Management Engineering, University of Calabria, Via P. Bucci, cubo 45, 87036, Rende, Cosenza, Italy;
[6] AGRIS-Sardegna, Agricultural Research Agency of Sardinia, S.S. 291 Sassari-Fertilia, 07100, Bonassai, Sassari, Italy;
[7] Department of Biological, Geological and Environmental Sciences, University of Catania, Corso Italia, 57, 95129, Catania, Italy;

*Correspondence to*: Adriano Guido (adriano.guido@unical.it)

**Abstract.** The coralligenous build-ups located in Mediterranean shelf in front of Marzamemi (SE - Sicily, Italy) represent useful natural examples to study the relationship between skeletal organisms and non-skeletal components in marine bioconstructions. Coralligenous build-ups are formed in open marine systems and their comparison with coeval bioconstructions (biostalactites) of confined environments, like submarine caves, allows depicting the complex interactions between metazoans and microbial communities in the formations of recent bioconstructions in different Mediterranean settings. In this study, two coralligenous build-ups were characterized in terms of organisms and sediments involved in their formation. The framework mainly consists of coralline algae and subordinate bryozoans and serpulids. Sponges affect the general morphology of the bioconstructions both interacting with skeletonised organisms and through bioerosion activity. The micrite or microcrystalline calcite is present in minor amount than other components that form the build-ups and consists of two types: autochthonous (*in situ*) and allochthonous (detrital). Fine autochthonous micrite mineralized directly inside the framework cavities and shows aphanitic or peloidal fabric, produced by organomineralization processes of soft sponge tissues and microbial metabolic activity, respectively. The detrital micrite occurring inside cavities derives from external sources or erosion processes of the bioconstructions themselves. This component has been classified as organic or inorganic based on the organic matter contents deduced by UV-Epifluorescence. A great amount of sponge live in cavities of the coralligenous build-ups and compete with carbonatogenic bacteria for the same cryptic spaces limiting the production of microbialites. The sharing of a similar relationship between sponges and microbial communities by coralligenous concretion and biotic crusts of particular submarine caves suggests that this competition is not habitat-specific. On the contrary, it may develop in a range of environmental settings, from open to cryptic systems, and could be used to clarify the role of metazoans *vs* microbialites in palaeoecological reconstructions.

Keywords: Coralligenous reefs; Sponges; Micrites; Geobiology; Mediterranean.



## 1. Introduction

Bioconstructions consisting of in-place reef-building organisms in temperate waters of the Mediterrenean Sea shelf are known as Coralligenous (Pérès & Picard, 1964). These structures are primarily made-up of calcareous red algae, which are able to develop algal-dominated frameworks peculiar for this basin (*e.g.*, Ballesteros, 2006). Pérès and Picard (1964) consider the Coralligenous a climax biocoenosis of the circalittoral plan, in which crustose coralline algae (CCA) and mineralized Peyssonneliales develop on primary or secondary hard bottoms, in dim-light conditions. Due to its importance as hot spot of biodiversity the European Community consider the Coralligenous among the most important habitats to be monitored and protected (see: Ballesteros, 2006; Gennaro et al., 2020), considering also its low accretion rate (0.06÷0.27 mm per year) (Sartoretto et al. 1996; Di Geronimo et al., 2001; Bertolino et al., 2019; Basso et al. 2022). Coralligenous is usually considered to be an association of heterogeneous communities (Ballesteros, 2006; La Rivière et al., 2021). Indeed, the external surface and the cavities of the build-ups host a rich association of calcareous red algae, sponges, bryozoans, serpulids, molluscs, and corals (Pérès, 1982; Bellan-Santini et al., 1994; Di Geronimo et al., 2002; Ballesteros, 2006; Rosso and Sanfilippo, 2009; Bertolino et al., 2017, 2019; Costa et al., 2019; Basso et al., 2022; Bracchi et al., 2022; Cipriani et al., 2023). Coralligenous build-ups: (i) modify the seafloor and the seascape (Laborel, 1961; Basso et al., 2007; Bracchi et al., 2015, 2017), (ii) promote the production of carbonate (Marchese et al., 2020), and (iii) may be recognized in the sedimentary succession (Bosence and Pedley 1982; Carannante and Simone 1996; Basso et al., 2007, 2009; Titschack et al., 2008; Bracchi et al., 2014, 2016, 2019). The presence of the coralligenous 3D structure and the related high biodiversity and biomass also determine the increase of the available resources. These attract microorganisms such as ostracods and foraminifera which, while not contributing directly to the bioconstruction, raise its biodiversity (e.g., Hong, 1982; Ballesteros, 2006; Sciuto et al., 2023).

Unlike other aspects, geobiological features of the Coralligenous and the role of skeletonised and non-skeletonised (*e.g.,* bacteria) communities in forming these build-ups have so far not been explored in detail. The lack of this information produces a gap between the knowledge of the build-ups developed in the open settings of the Mediterranean Sea and those forming in confined environments, like the "biostalactites" of submarine caves. Actually, "biostalactites" from Apulia, Adriatic Sea (Onorato et al., 2003; Belmonte et al., 2009; Rosso et al., 2020; Guido et al., 2022), Sicily (Guido et al., 2012, 2017a; Sanfilippo et al., 2015), Lesvos Island, Aegean Sea (Sanfilippo et al., 2017; Guido et al., 2019a, 2019b); and Cyprus, Levantine Sea (Guido et al., 2017b; Jimenez et al., 2019) have been studied in detail and the biotic and abiotic processes involved in their formation have been clarified. These studies showed the fundamental role of bacteria in strengthening the bioconstructions, through the biomineralization processes of autochthonous micrite (Guido et al., 2013, Gischler et al., 2017a).

Like these systems, it is conceivable that also the high porous framework of the Coralligenous could promote the development of non-skeletal biomineralization processes. The biomineralization term indicates mineralization processes associated to biotic activity that have been extensively investigated mainly in carbonate rocks (Riding, 2000, 2011; Van Driessche et al., 2019). The crystal nucleation can be: (i) controlled directly by the organisms; (ii) induced by microbial communities; or (iii) influenced by the presence of cell surface organic matter (Lowenstam & Weiner 1989; Benzerara et al., 2011; Phillips et al., 2013; Anbu et al., 2016; Riding & Virgone 2020). In all these cases, the formation of biominerals also depends on the chemical-physical conditions of the environment (Riding & Liang 2005; Riding 2011). To date no research on possible non-skeletal biomineralization processes has been addressed for the Coralligenous. In the frame of the project "CRESCIBLUREEF", and based on two build-ups collected in south-eastern Sicily (Ionian Sea, Italy) off the Marzamemi village, we aim to improve knowledge on the Coralligenous: (1) describing the origin of the sediments filling the cavities of the skeletal framework; (2) investigating the relationships between sponges and microbial processes





through biomineralization-mediated processes; and (3) comparing the Coralligenous, formed in open marine systems, with the bioconstructions of submarine caves developed in confined marine settings.

## 2. Materials and methods

In the studied area coralligenous bioconstructions, mostly columnar-shaped and distributed in more or less dense clusters, extend widely in a belt between ca. 36 and 100 m depth. The image analysis and computed axial tomography reveals that the bioconstruction's framework is mainly formed of coralline algae and in minor amount of invertebrates (mostly serpulids and bryozoans) and cavities filled with sediment (Bracchi et al., 2022; Varzi et al., 2023). The surfaces of the studied build-ups were covered with a dense, up to 8–10 cm thick, canopy of fleshy algae and locally, also by subordinate erect bryozoan colonies (Bracchi et al., 2022; Donato et al., 2022; Rosso et al., 2022, 2023; Sciuto et al., 2023). The most abundant components are CCA and mineralized Peyssonneliales with a cover of up to about 61%. Fleshy algae are also abundant, up to about 32%. The faunal groups have negligible covers, except for bryozoans reaching about 12% (Bracchi et al., 2022).

Two build-ups called CBR2_3_7c and CBR2_4_21c from this coralligenous field located in the Ionian Sea were collected (Fig. 1). The build-up CBR2_3_7c (36° 43.394′ N; 15°09.469′ E) was sampled at 37 m depth in a zone covered by coralligenous hybrid banks (Bracchi et al., 2017), made of distinct, though sometimes coalescent, coralligenous columnar build-ups (Fig. 1B and D). The build-up CBR2_4_21c (36°43.454′ N; 15°09.657′ E) was collected at 36 m depth in an area characterized by sparse and distinct build-ups growing on biogenic gravel and sand substrate (Fig. 1C and E).


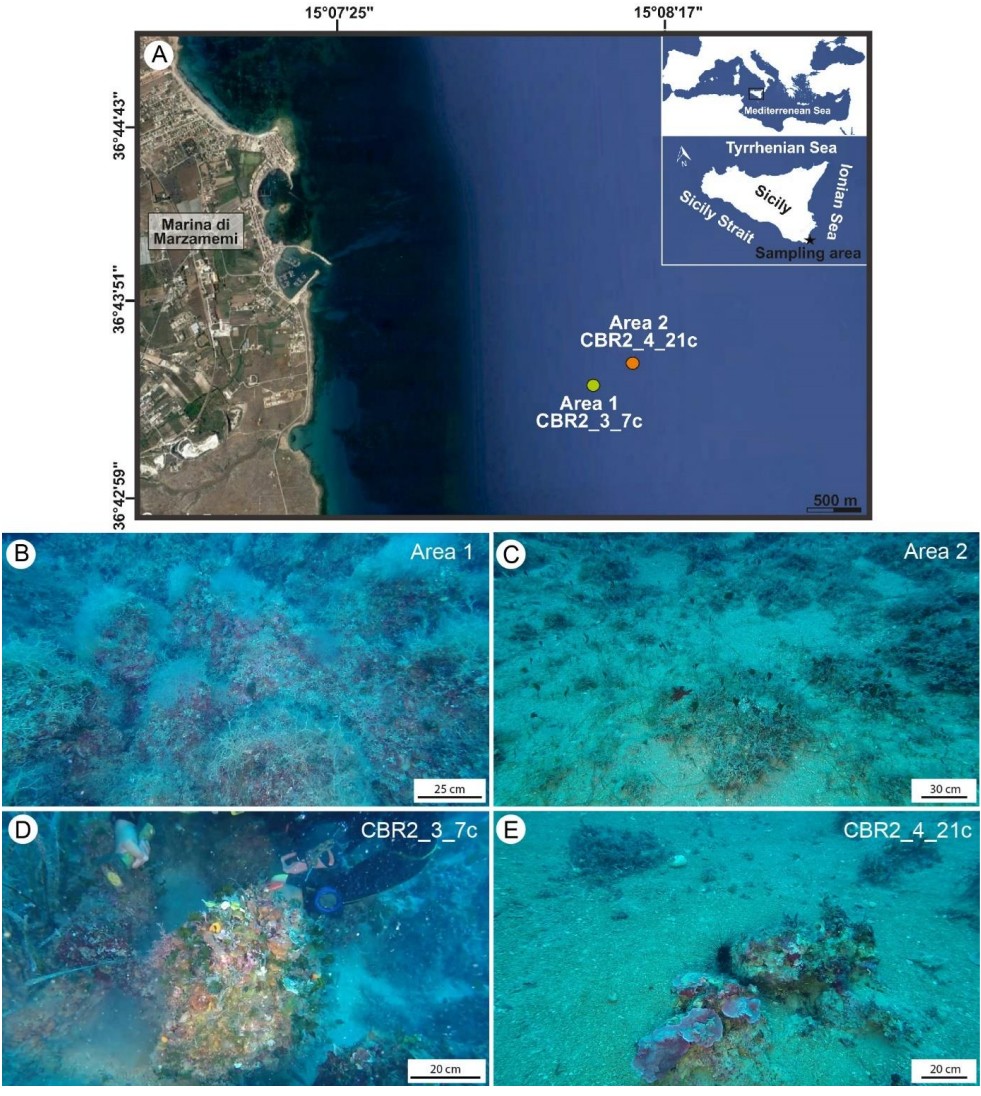

99

**Figure 1:** (A) Location of the study areas off the coast of Marzamemi village in the Ionian Sea. (B-C) Underwater photos of the sampling sites; (B) Area 1, characterized by high coralligenous cover, from where the CBR2_3_7c build-up was collected at 37 m.

After drying, the build-ups were cut with a diamond saw following the putative grow direction of the structures. The cutting planes showed that the coralline algae framework forms a highly porous structure with cavities filled with sediments (Fig. 2C-H). Photos at macroscale were acquired at the University of Milano-Bicocca with a Nikon D3500 camera. Photos at mesoscale were acquired at the University of Catania through a stereomicroscope Zeiss Discovery V8A stocked with an Axiocam MRC and a system for automatic acquisition of the images (Axiovision). To investigate the role of sediment in the growth and stabilization of the skeletal components (at micro and nano-scale) small fragments (Fig. 2G and H) and thin sections (Fig. 3) were analysed at the Laboratory of Geobiology of the Department of Biology, Ecology and Earth Science, University of Calabria.
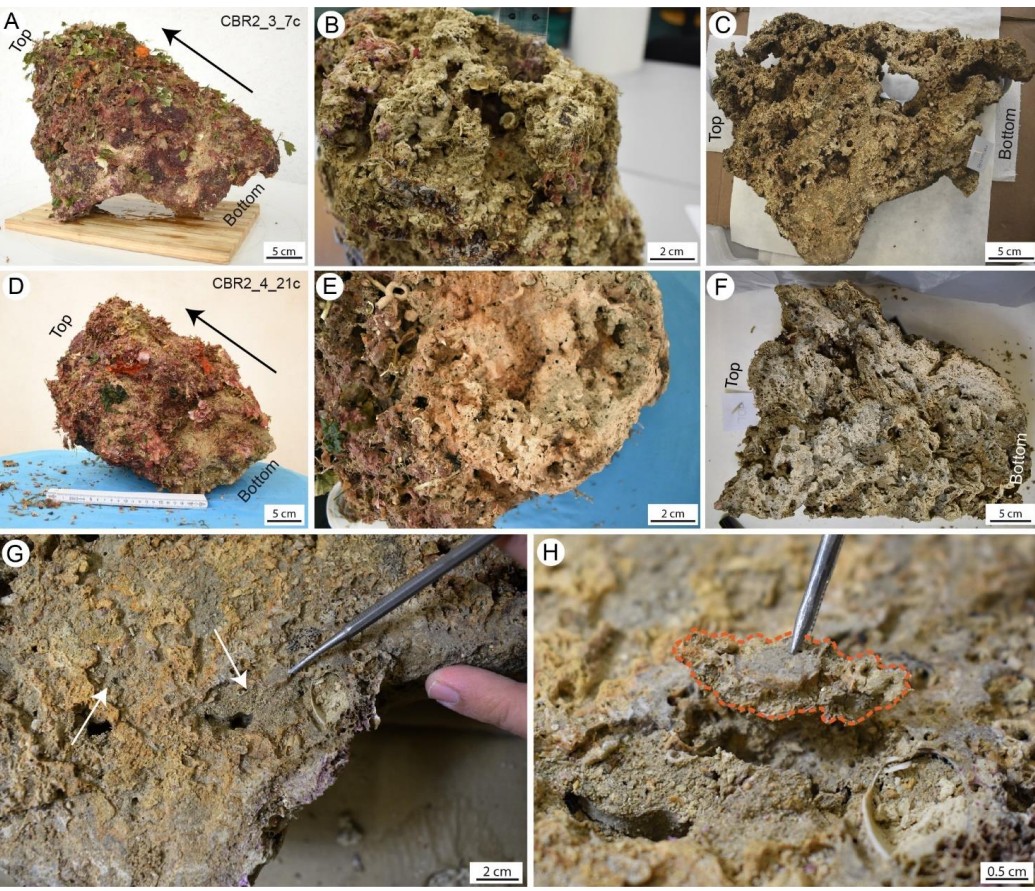

111

**Figure 2:** (A-D) Sampled coralligenous build-ups; note the high biogenic cover (largely consisting of soft-bodied
organisms) on the external surfaces. Black arrows indicate the main growth direction. (B and E) Surfaces of detachment
from the sea-bottom. (C and F) plane slab after the longitudinal cut of the build-ups showing the internal structure; note
the high porous skeletal framework with cavities partly filled with sediment. (G and H) Selection and sampling of small
micritic fragments from the cutting surfaces; (G) The white arrows point to cavities partly filled with sediment on a cutting
plane of the CBR2_3_7c build-up; (H) detail of a fragment (red dotted line) sampled for the analyses. Bottom indicates
the portion of build-ups detached from the substrate. A, C, and H from Cipriani et al. (2023).

A total of twenty-nine small blocks, selected following a grid with sides of 5x3 cm on the cutting plane, have been utilized
for thin sections preparation (Fig. 3). The blocks have been chosen based on the mesoscopic aspect and relative amount
of skeletal components and micrite sediments. The fragments and thin sections have been investigated using an optical
microscope (Zeiss Axioplan Imaging II) at different magnifications (2.5x, 5x, 10x, 20x, and 40x). Thin sections were
used for point counting analyses of the main components (skeletons, micrite and cavities). A total of 300 points per thin
section were counted. Fluorescence intensity has been evaluated in incident light utilizing a Hg high-pressure vapour bulb
and high-performance wide bandpass filters (band-pass filter 436/10 nm/long-pass filter 470 nm, no 488006, for the green
light; and band-pass filter 450–490 nm/long-pass filter 515 nm, no. 488009, for the yellow light). UV-epifluorescence
was used to discriminate the presence and distribution of organic compounds and to recognize biotic and abiotic
components, especially in those cases showing a similar general aspect under reflected light.
Selected fragments, used for Scanning Electron Microscopy (SEM) observations and microanalysis, were carbon coated.
The SEM apparatus was used is Ultra High Resolution(UHR-SEM) – ZEISS CrossBeam 350 with the following





condition: resolution 123 eV, high voltage 10 keV, probe current 100 pA and working distance 11 mm. Mineralogical
and chemical compositions were investigated under high voltage 15 keV, probe current 60 mm, working distance 12 mm,
take-off angle 40° and, live time 30 sec.

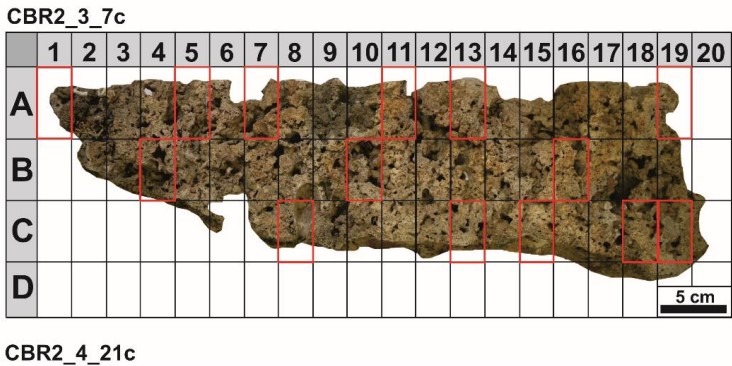

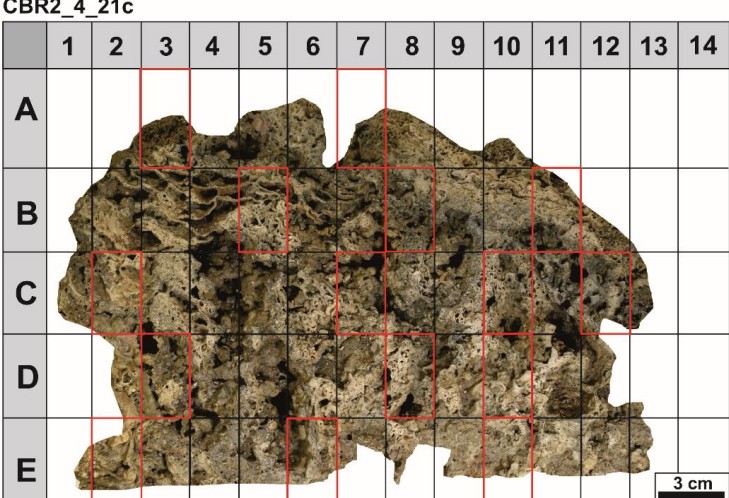


**Figure 3:** Cutting plane of the coralligenous build-ups with the superimposed grids of 5x3cm sized rectangles. Cells
selected for thin sections are rimmed in red.

**3. Results**
**3.1 Morphology and framework at mesoscale observation**
The CBR2_3_7c build-up is 56 cm high, with circumferences of 59 cm at the base, 78 cm at the top and a maximum of
116 cm in between (Fig. 2A). The CBR2_4_21c build-up is 38 cm high, with circumferences of 71 cm at the base, 52.5
cm at the top and a maximum of 112 cm in between (Fig. 2D). Both build-ups show a prevalent upward growth and a
rough surface.
The surfaces of detachment (Fig. 2B and E, for CBR2_3_7c and CBR2_4_21c, respectively) and selected longitudinal
plane slabs (Fig. 2C and F, for CBR2_3_7c and CBR2_4_21c, respectively) highlight the internal framework of the two
build-ups, characterized by a primary skeletal framework forming a high porous structure, with cavities ranging from few
millimeters to ten centimeters. Two types of cavities are present: primary and secondary. Primary cavities represent the
interspaces generated during the superimposition of skeletonized encrusting organisms. Their surfaces are sometime




encrusted by skeletonized organisms, mainly serpulids and bryozoans. The secondary cavities could derive from the
necrolysis of soft-bodies organisms sandwiched between the skeletonised ones or from the bioerosion produced by
endolytic organisms, mainly sponges and subordinately bivalves. Cavities show cylindrical-barrel shapes (like those
produced by boring bivalves) or irregular shapes and may be empty or (partially or totally) filled with sediment. This
appears either brownish-greyish or greenish. brown to grey/dark grey in colour. Brownish-greyinsh sediment is muddy,
usually located in larger cavities, and includes a bioclastic component (planktonic and benthic foraminifer shells, small
fragments of coralline algae, serpulids, bryozoans, ostracods, molluscs) and appears loose. Greenish coloured sediment
is subordinate and distributed in smaller (millimetre- to centimetre- sized) cavities; it seems cemented and consists of
mud lacking skeletal fragments, at mesoscale observation (Fig. 2G). This component mainly occurs along the borders of
partly filled cavities possibly originally occupied by sponges, as testified by remains of their soft tissue connected with
spicules (Fig. 4A and B). The two types of sediment (loose and cemented) seem distributed according to the size of the
cavities but they do not show a preferential distribution, from the bottom to the top of the structures, inside the build-up
frameworks.

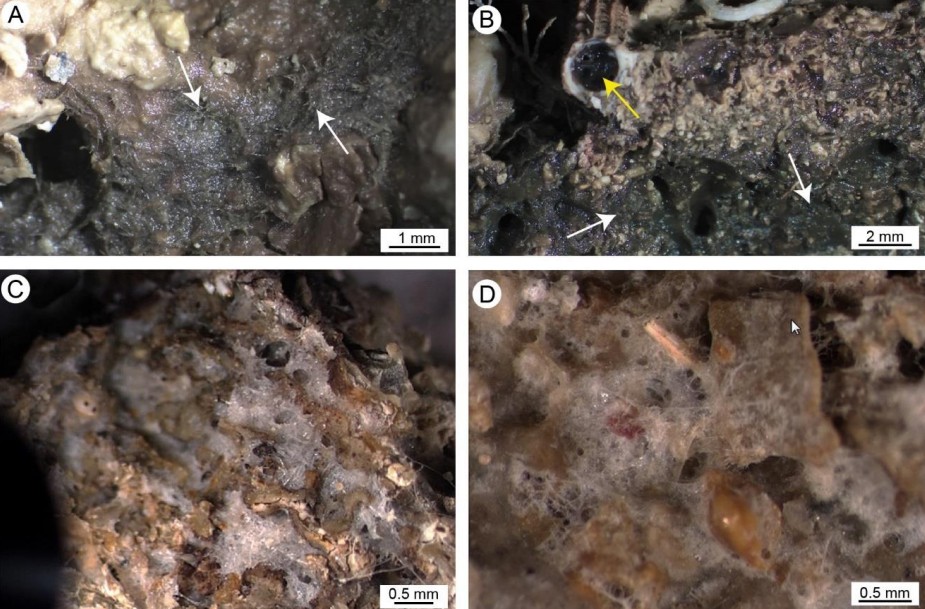


**Figure 4:** (A-B) Fine micritic sediment associated to remain of sponge tissue (white arrows) in internal cavities of the
framework; the yellow arrow points to a sponge colonizing the internal cavity of a serpulid tube. (C and D) Pervasive
colonization of sponges in internal cavities and the external surface, respectively.
**3.2 Analyses of the fragments**
The observation of the build-up fragments at the microscale (Fig. 2H and Fig. 5) highlight the superimposition of
successive generations of different taxa producing the crusts. The skeletal framework is mainly composed of CCA (Fig.
5A-C). Serpulids (Fig. 5A, D and E) and bryozoans (Fig. 5A, F and G) participate subordinately to the formations of the
structure. Sponges seem to concur to the general morphologies, regulating the direction of growth of the encrusting
organisms, and altering the internal body of the build-ups through bio-erosive processes. The activity of sponges is evident
both on external surface and internal microcavities of the fragments testifying the pervasive colonization of these
organisms at different scales (Fig. 4). Spicules sometimes are associated with fine carbonate mud (Fig. 4A and B), in





other cases their original presence in empty cavities is testified by organic remains with spicules (Fig. 4C and D) and
specific micro-morphologies of the cavity boundaries which testify the boring activity of these organisms. Molluscs and
other skeletonized invertebrates make a negligible contribution to the build-up growth. Rare solitary corals are also
present.

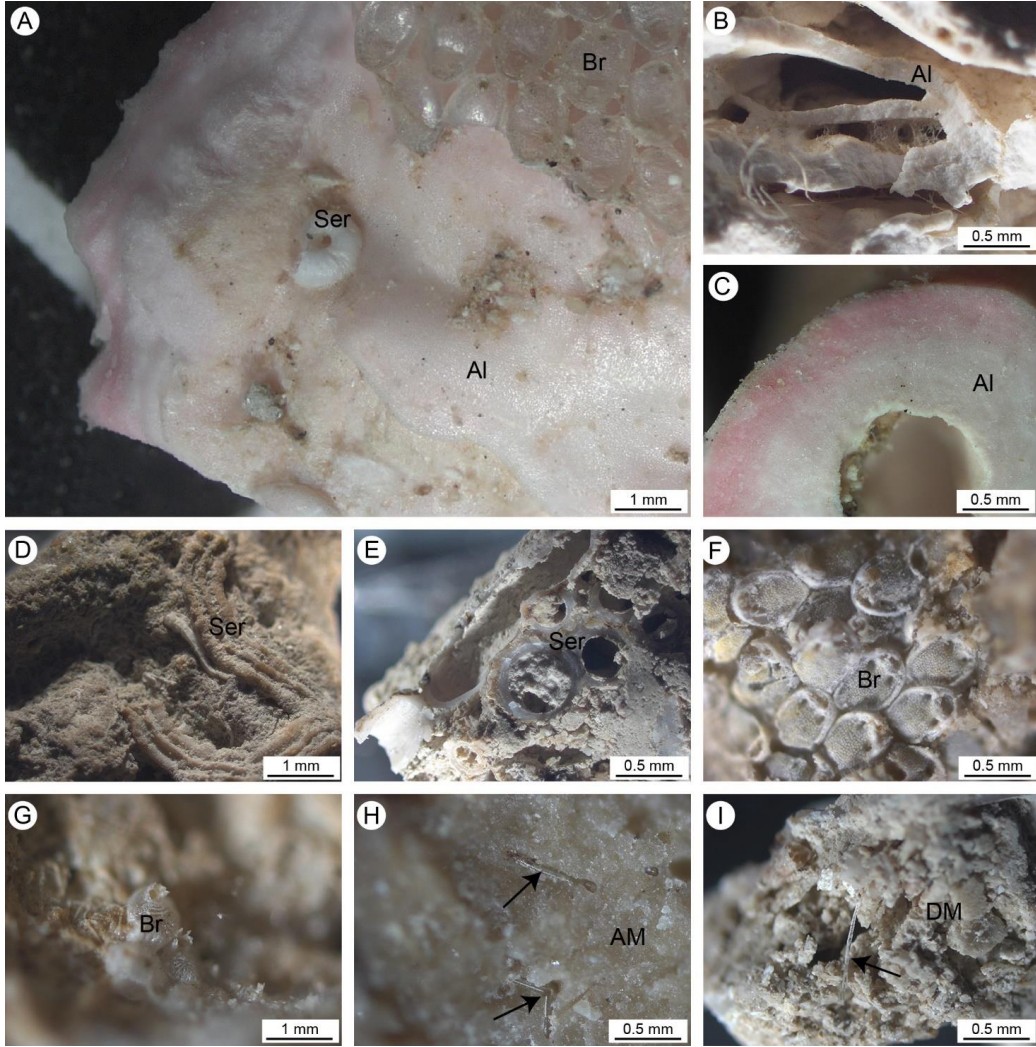


**Figure 5:** (A-B) Main carbonate components of the small fragments collected on the cut surfaces of the build-ups. (A)
Pink crustose coralline algae (Al) encrusted by serpulids (Ser) and bryozoans (Br). (B) Different generation of crustose
coralline algae encrusted one on the top of the others. (C) Cross section of a crustose coralline alga. (D) Serpulid tube
encrusting cemented micrite. (E) Section of serpulid tubes intermingled with micritic sediments. (F and G) bryozoan
colonies. (H) Dense and homogeneous autochthonous micrite (AM) engulfing sponge spicules. (I) Heterogeneous and
loose, detrital micrite (DM) sediment engulfing sponge spicules. Black arrows point to sponge spicules engulfed in the
micrite sediments.
Two types of sediments (micrite) have been observed: homogeneous or autochthonous (compact and dense; Fig. 5H) that
emits a bright fluorescence when excited with UV-light, and heterogeneous or detrital (less cemented and rich of fine
bioclasts; Fig. 5I) that does not emit fluorescence (Fig. 6). Both types of micrites engulf sponge spicules.
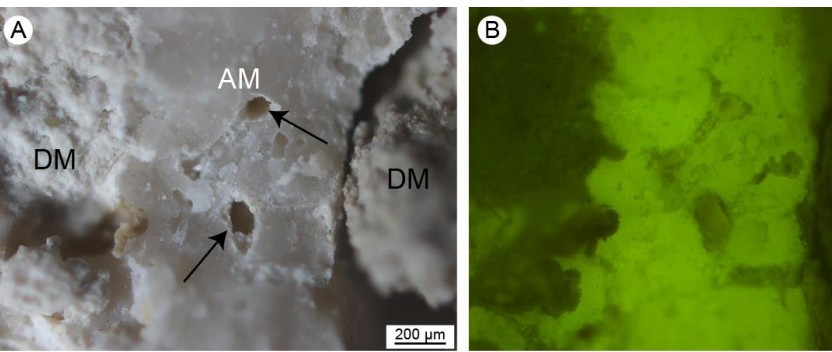


**Figure 6:** Micrite sediments from CBR2_4_21c build-up observed in reflected light (A) and ultraviolet light (B). The bright fluorescence indicates a high content in organic matter of the autochthonous micrite, whereas the absence of fluorescence of the detrital micrite denotes an inorganic origin. Note the microcavities left by sponge spicules (black arrows). AM: autochthonous micrite; DM: detrital micrite.

**3.3 Microfacies characterization**

Thin sections observation confirms the main role of skeletonized organisms in forming the carbonate framework of the Coralligenous (Fig. 7). CCA are the main builders through successive generations of specimens encrusted one on top of the others (Fig. 7). Bryozoans and serpulids play a subordinate bio-constructional role. Sponges are abundant and their amorphous material remains are widely distributed in cavities and microcavities, often associated to lose sediments, engulfed in cemented micrite. Numerous other bioclasts, produced by organisms that do not participate directly to build the bioconstructions were detected in the cavities together with muddy materials. Non-skeletal carbonate material was also detected (see below).




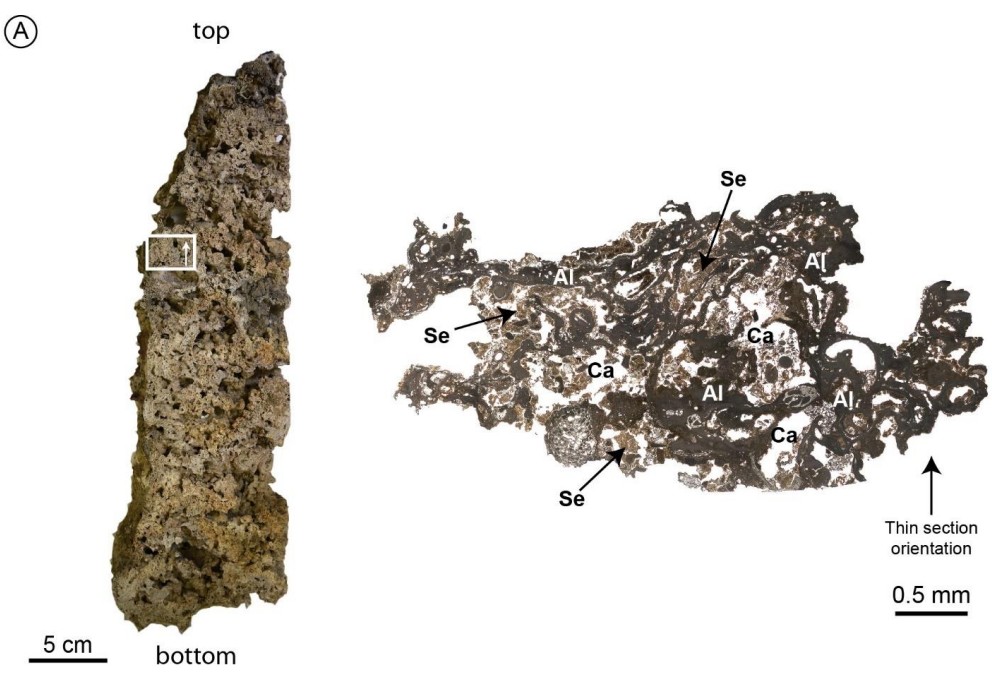

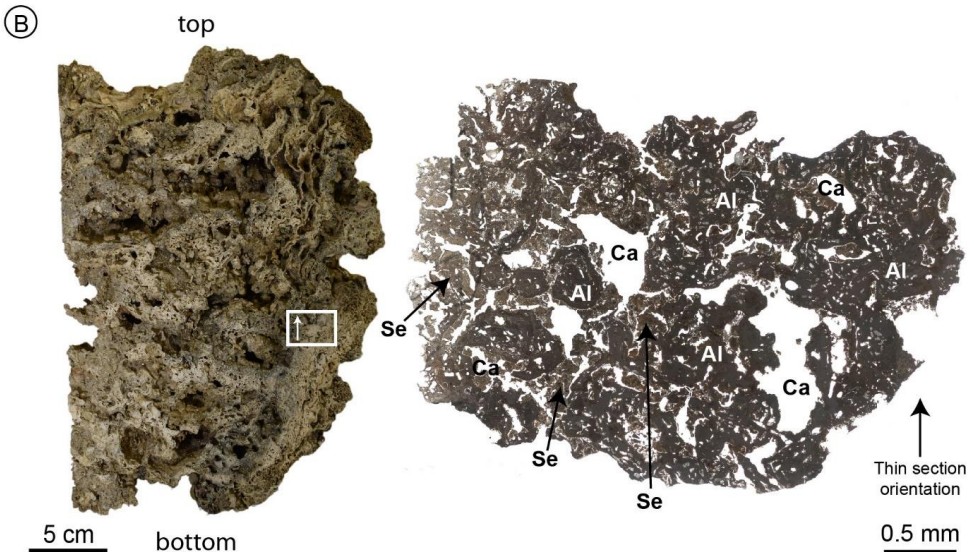

**Figure 7:** Cutting planes (left) and optical microscope photocomposition of a representative thin sections (right) of the build-ups CBR2_3_7c (A) and CBR2_4_21c (B). Note the primary role of the coralline algae (Al) as major constituent of both build-ups and the presence of cavities (Ca) empty or filled by micrite sediments (Se). White rectangles and the arrows within in indicate the location and the orientation of the thin sections.

### 3.3.1 Skeletal components

Articulate and crustose coralline red algae are the main skeletal component detected in all thin sections (Fig. 8). The skeletons of the algae clearly form a continuous framework at the mesoscale (Fig. 7) but at the microscale, laminae are



rarely continuous, often showing traces of bioerosion (Fig. 8A-C, F). Two main types of bioturbations are distinguishable:
(i) irregular borings and (ii) tube-like micro-borings. Borings, showing sizes from a few microns to a few millimeters are
mainly formed by endolithic sponges that corrode and perforate skeletons. These cavities could be: a) empty due the
decaying of the sponge's organic tissue (Fig. 8B); b) filled with detrital sediment (Fig. 8B); or c) filled with spicules and
remains of organic matter deriving from soft tissue decay of the boring sponge *Cliona* sp. (Fig. 8F).
The skeletons often show also micritization phenomena contributing to the alteration of the original microstructures
together with bioerosion.

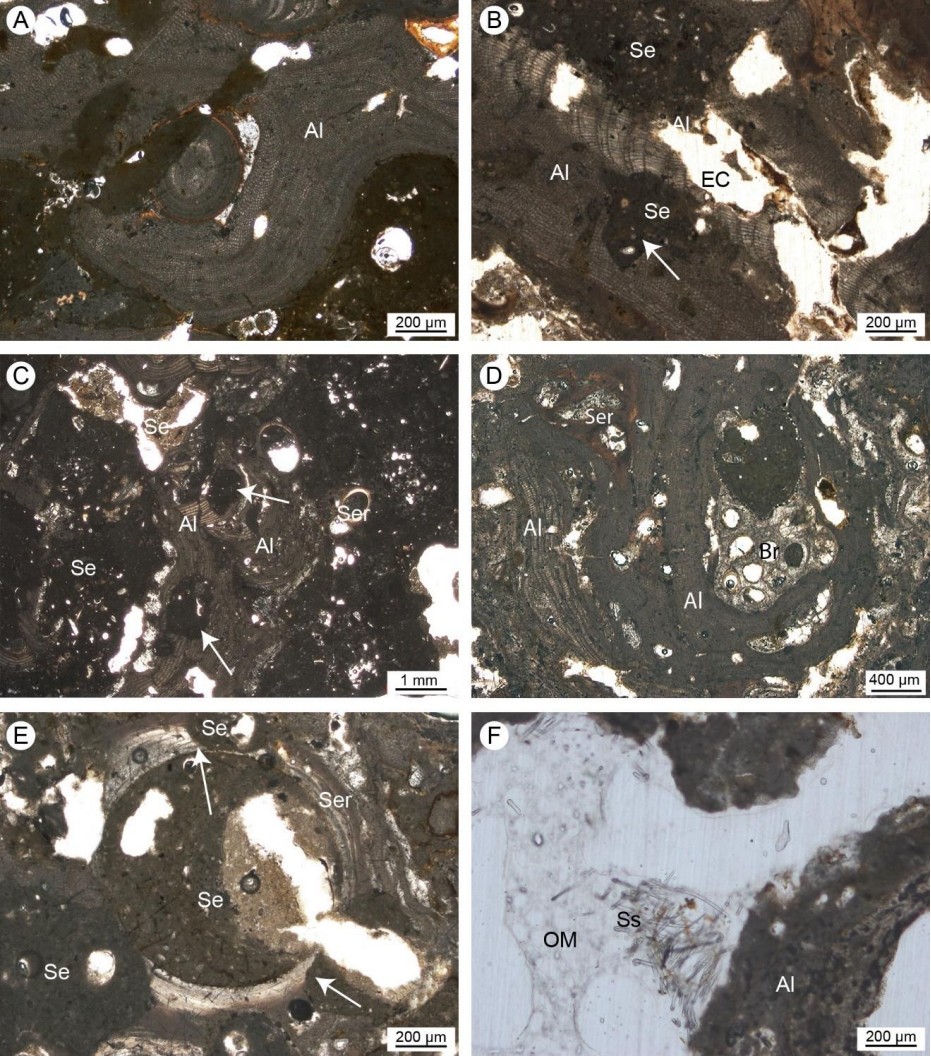

**Figure 8:** (A-C) Skeletal tissues of algae (Al) showing microcavities due to bioturbations (white arrows); the cavities are empty (EC) or filled with sediment (Se). (D) Strict interconnection among the main builders of the studied coralligenous build-ups (Al: algae; Ser: serpulids; Br: ? bryozoans). (E) Bioeroded serpulid skeleton (Ser); note the borings (white arrows) filled with sediments (Se). (F) Bioerosion cavity in algae (Al) with remains of organic matter (OM) and spicules (Ss) derived from sponge's decay (Cliona sp.). [C-D : CBR_2_4_21c; A-B, E- F: CBR_2_3_7c].

The biological activity involved in these bioerosional processes was not investigated but could represent a further step in
the reconstruction of the complex biological relationships which develop in these coralligenous ecosystem. The erosive
action of sponges is clearly visible where remains of amorphous material and spicules are associated to corroded substrates
(Fig. 9).
The sponge spicules, mainly belonging to species of the genus *Jaspis*, also fill the algae's conceptacles and often, small
spherical corpuscles are recognizable among spicules in UV- epifluorescence.

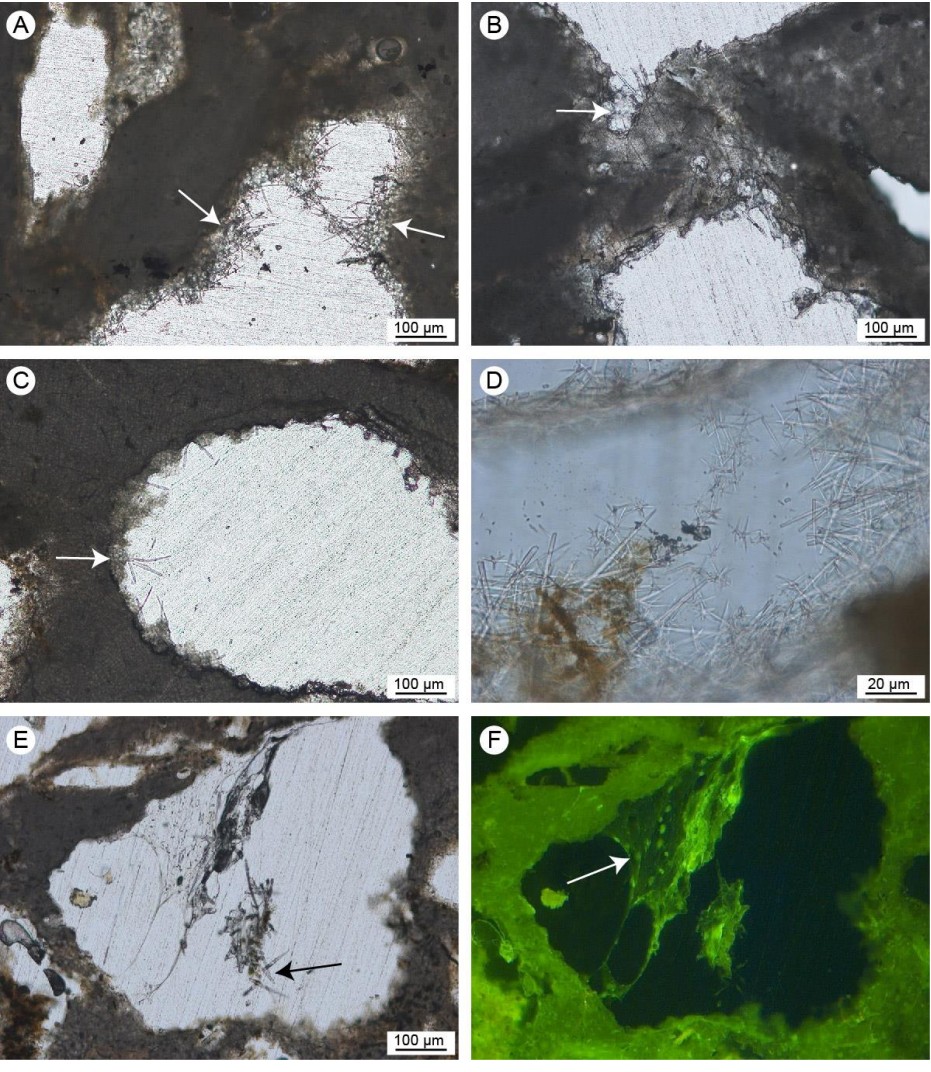


**Figure 9:** (A-C) Microcavities inside the skeletal framework of the coralligenous build-ups showing sponge spicules
(plurispicular tracts) associated to the corroded substrates (white arrows). (D-F) Details of sponge spicules of Jaspis sp.
(D) and Cliona sp. (E) observed in transmitted light, and UV-epifluorescence (F); the non-fluorescent siliceous spicules
are engulfed in remains of organic matter (F) deriving from sponge soft tissue decay. [A-C: CBR_2_4_21c; D-F:
CBR_2_3_7c].
Serpulids are common in the microfacies of the two build-ups. They occur as isolated or grouped tubes with outer
diameters ranging from 200 µm to 1 mm. The number of clustered tubes usually increases with decreasing of their sizes.





They are empty, or filled with sediment, or sometimes occupied by endolytic sponges (Fig. 4B). Serpulids are often
encrusted or encrust bryozoans and/or algae (Fig. 8D). Serpulid species tentatively recognized in thin sections belong to
*Serpula lobiancoi* at different degree of preservation. The original microstructure is observable in the bigger ones, while
dissolution, recrystallization and/or micritization often alter the smaller tubes. Serpulids suffer also bioerosion by
endolytic organisms (Fig. 8E).
Remains of amorphous material and spicules inside the skeletons of bryozoans may derive from insinuating or bioeroding
sponges, like observed for other skeletonised taxa.

### 3.3.3 Non-skeletal carbonate components: autochthonous and allochthonous (detrital) micrite

The cavities of the skeletal framework are filled by different micrite types. These sediments were distinguished under
light microscope examination and UV-Epifluorescence. The texture and organic matter content allowed to distinguish an
autochthonous and an allochthonous (detrital) micrite.
The autochthonous micrite consists of very fine-grained calcite and shows aphanitic (Fig. 10A-F) or peloidal to clotted
peloidal textures (Fig. 10G-H). The autochthonous aphanitic micrite displays a light brown colour and shows a
structureless mud-supported texture with rare bioclasts. Autochthonous peloidal micrite displays a darker colour, do not
exhibit grain-supported textures and shows interclot areas indicating a not gravitational genesis. Peloidal micrite fills
microcavities or coats serpulid tubes or other bioclasts. Peloids aggregate often in clots separated by calcite microspar
(euhedral Mg–calcite crystals), forming a clotted texture or, less commonly, a compact texture through the coalescence
of several clots. Peloidal and aphanitic microfabric derive from mineralization mediated indirectly by organic processes
and represent *in situ* precipitation of the micrite, whose syndepositional cementation contributes to stabilize the skeletal
structures of the build-ups.
Aphanitic and peloidal micrites show a bright autofluorescence under UV-light indicating a high content of organic matter
relicts, most likely related to the bio-induced crystal precipitation. Aphanitic micrite is widely associated with the presence
of sponge spicules (Fig. 10A-F), and generally fills bioeroded cavities inside the skeletal framework. On the contrary,
peloidal micrite, is not associated to sponge spicules and often occludes serpulid tubes and spaces within adjacent
individuals, contributing at cementing the skeletons together. Small terebellid tubes are often associated with this micrite
type (Fig. 10G-H).
The amount of the autochthonous micrite is variable in the thin sections but always represents a minor component in
comparison to the skeletal framework, and shows a different distribution along the bottom-top direction of each build-up,
with the major content in the column CBR_2_4_21c.






**Figure 10:** (A-F) Aphanitic autochthonous micrite (AAM) engulfing sponge spicules (white arrows) observed in
transmitted light (left) and UV-epifluorescence (right); the bright epifluorescence of the AAM indicates the presence of
organic matter relics closely related to the bioinduced crystals. (G-H) Peloidal autochthonous micrite (PAM) engulfing
some agglutinated skeletons of terebellids (Te) observed in transmitted light (left) and UV-epifluorescence (right); even





in this case the bright epifluorescence of the PAM indicates the presence of organic matter relics closely related to the
bioinduced crystals. [A-B, E-H: CBR_2_4_21c; C-D: CBR_2_3_7c].
The detrital micrite shows a light brownish colour and is characterized by a texture with variable density (Fig. 11). Two
types of detrital micrite with different textures have been tentatively recognised: organic and inorganic. Organic detrital
micrite shows a denser muddy texture and it is enriched in bioclasts, intraclasts and sponge spicules (Fig. 11A, D, E). It
shows a very faint to scarce epifluorescence (Fig. 11F). The inorganic detrital micrite is made up of particles with larger
size (in the silty range) and includes a minor amount of bioclasts. Due to the absence of epifluorescence under UV-light
it is assumed an inorganic nature of these components. Detrital micrites represent the main non-biomineralized component
of both build-ups, and fill primary, inter- and intra-skeletal cavities, and secondary micro-cavities generated by boring
organisms.

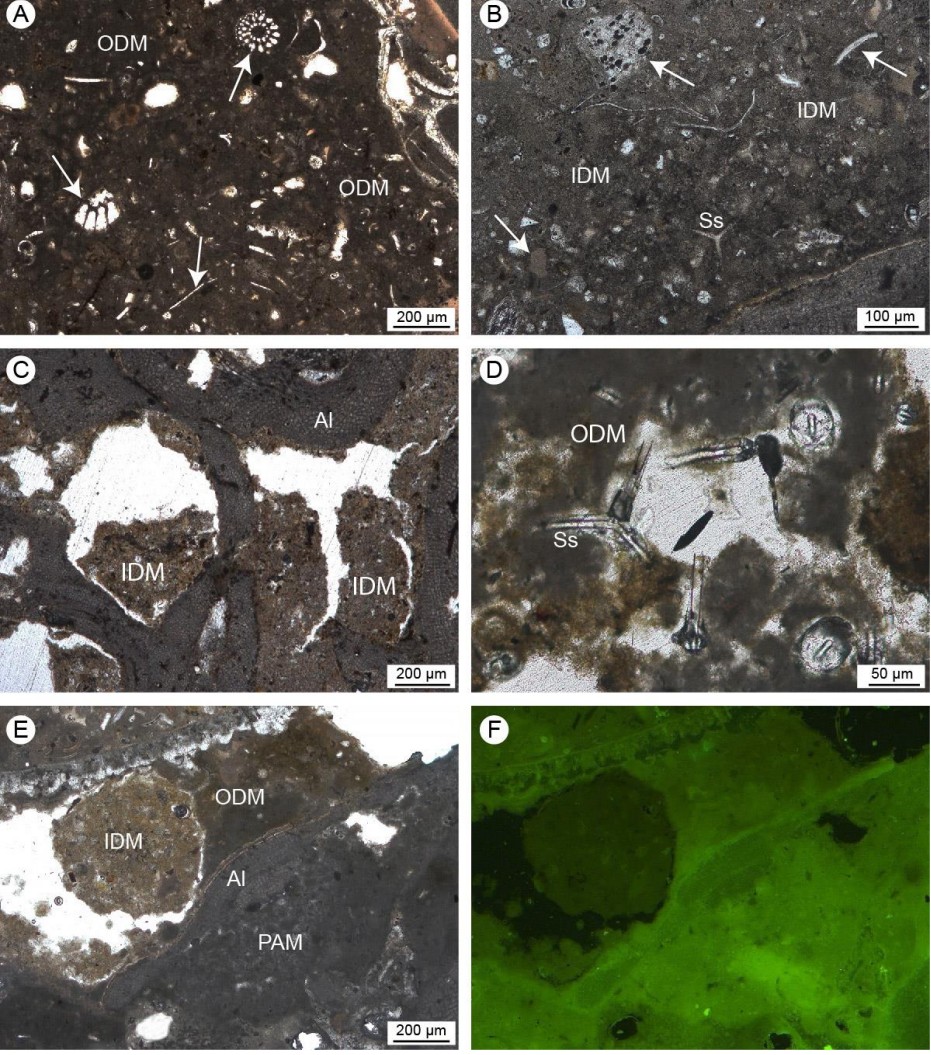


**Figure 11:** Detrital micrite textures. (A) Organic detrital micrite (ODM) with a dense muddy texture enriched in bioclasts,
intraclasts and sponge spicules (white arrows). (B) Inorganic detrital micrite (IDM) consists of particles with larger sizes
incorporating a minor amount of bioclasts (white arrows). (C) Microcavities in bioturbated algae (Al) filled with IDM.





(D) ODM engulfing sponge spicules (Ss). (E-F) Relationship between IDM (light brown), ODM (dark brown), algae and
peloidal autochthonous micrite (PAM) observed in transmitted light (left) and UV-epifluorescence (right); faint and scarce
fluorescence of the detrital micrite denotes an inorganic nature of these components. [A-B, D-F: CBR_2_4_21c; C:
CBR_2_3_7c].

### 3.3.4 Cements

Cement only sporadically fills cavities representing a subordinate component of the build-ups. Two types of cement have
been recognised: primary (syndepositional) and secondary (diagenetic) cement. Syndepostional cement shows
isopachous, botryoidal and peloidal microcrystalline fabric. Isopachous cements develop with homogeneous fringes on
the surface of intra- and inter-skeletal microcavities. Botryoids consist of dome-shaped hemispheres built by radiating
fibrous calcite crystals and crystal fans filling the primary cavities and voids created by bioerosion processes. Peloidal
microcrystalline cement is composed of tiny peloids within a microcrystalline calcite matrix; it was mainly detected in
small intra-skeletal cavities. Secondary cement is rare and fills residual microcavities with drusy micro-textures.

### 3.3.5 Point counting analyses

Seven structural and non-structural components have been counted: carbonate framework builders, bioclasts,
autochthonous micrite, organic and inorganic detrital micrite, boring sponges and empty cavities. Bioclasts include all
skeletal remains of organisms that do not participate to the formation of the skeletal framework but were trapped inside
the cavities either first living in association with the build-up or transported by neighbouring habitats. Among these,
bivalves, gastropods, foraminifers, ostracods, echinoid plates and spines, and algal fragments have been recognized. The
boring sponges counting class includes perforations interpreted as originally occupied by sponges, because infilled of
amorphous organic remains rich in spicules. It is worth to note that this component could be underestimated due to the
cutting procedures and the preparation of the thin sections which could have washed away the residues of the sponge
tissue that originally occupied the cavities. The analysis shows the following average percentages for the CBR_2_3_7c
and CBR_2_4_21c build-up respectively: 46.2% and 47.1% carbonate framework builders (algae, serpulids, bryozoans);
4.4% and 3.6% other bioclasts (planktonic and benthic foraminifer shells, ostracods, molluscs, echinoid plates and spines
and, small fragments of coralline algae); 3.6% and 9.2% autochthonous micrite, 16.4% and 14.3% organic detrital micrite,
13.6% and 9.3% inorganic detrital micrite, 14.8% and 15.4% empty cavities and, 1% and 1.1% boring sponges (Tables
310  1-2).





**Table 1.** Quantitative percentage of the main components recognised in the thin section samples from CBR_2_3_7c build-up.

| Samples Name | Carbonate framework builders (%) | Other bioclasts (%) | Autochthonous micrite (%) | Organic detrital micrite (%) | Inorganic detrital micrite (%) | Empty cavities (%) | Boring sponges (%) |
|---|---|---|---|---|---|---|---|
| A5 | 60.9 | 4.2 | 2.4 | 1.7 | 20.7 | 10.1 | 0 |
| A7 | 35.2 | 5.4 | 3.2 | 17.8 | 17.2 | 20.2 | 1 |
| A11 | 42.9 | 5.5 | 1.2 | 31 | 11.5 | 6.7 | 1.2 |
| A13 | 49,6 | 2.7 | 2.9 | 11 | 16.6 | 17.2 | 0 |
| A19 | 36.7 | 4.8 | 8.9 | 30.5 | 8.6 | 9.7 | 0.8 |
| B4 | 41.7 | 4 | 3.5 | 7.5 | 10.3 | 28.3 | 4.7 |
| B10 | 60,7 | 4.3 | 1.4 | 8.4 | 10.5 | 14.5 | 0.2 |
| B16 | 54 | 6 | 0 | 14.3 | 7.6 | 16.9 | 1.2 |
| B19 | 32,7 | 2.1 | 7.6 | 28.5 | 15.5 | 13.6 | 0 |
| C8 | 51 | 3.6 | 0.9 | 11.5 | 16.6 | 15.8 | 0.6 |
| C15 | 45,6 | 3.1 | 0.9 | 15.9 | 15.3 | 19.1 | 0.1 |
| C13 | 38.1 | 4.4 | 9.3 | 18.1 | 13.9 | 12.8 | 3.3 |
| C18 | 50.7 | 7 | 4 | 16.6 | 12.6 | 8.6 | 0.5 |
| Av. (%) | 46.2 | 4.4 | 3.6 | 16.4 | 13.6 | 14.8 | 1 |

**Table 2.** Quantitative percentage of the main components recognised in the thin section samples from CBR_2_4_21c build-up.

| Samples Name | Carbonate framework builders (%) | Other bioclasts (%) | Autochthonous micrite (%) | Organic detrital micrite (%) | Inorganic detrital micrite (%) | Empty cavities (%) | Boring sponges (%) |
|---|---|---|---|---|---|---|---|
| A7 | 40.8 | 5.3 | 9.6 | 22.2 | 10.3 | 11.5 | 0.3 |
| B5 | 55.9 | 2.2 | 9.7 | 3.1 | 9.9 | 19 | 0.2 |
| B8 | 59.4 | 3 | 3 | 6.2 | 10.9 | 15.8 | 1.7 |
| B11 | 58.1 | 2.9 | 5.1 | 5.9 | 6.9 | 20.4 | 0.7 |
| C2 | 28.9 | 4.4 | 14.7 | 20.7 | 8.7 | 21.4 | 1.2 |
| C7 | 57.8 | 1.2 | 7.6 | 9.3 | 5.3 | 15 | 3.8 |
| C10 | 42.7 | 2.5 | 7.3 | 25.5 | 2.7 | 18.3 | 1 |
| C12 | 49.1 | 3.3 | 6 | 3 | 13.3 | 24.3 | 1.1 |
| D3 | 48.1 | 1.6 | 8 | 22.2 | 8.6 | 11.1 | 0.4 |
| D8 | 42.2 | 5.2 | 15.5 | 13.6 | 12.7 | 10.7 | 0.1 |
| D10 | 54,6 | 2.8 | 7.8 | 11.9 | 11.2 | 11.7 | 0 |
| E2 | 32 | 10.2 | 19.7 | 13.1 | 9.7 | 14 | 1.3 |
| E6 | 37 | 1.6 | 10.1 | 27.2 | 11.5 | 12.6 | 0 |
| E10 | 53.8 | 4 | 5.3 | 16.4 | 8.8 | 10 | 1.7 |
| Av. (%) | 47.1 | 3.6 | 9.2 | 14.3 | 9.3 | 15.4 | 1.1 |

Among the non-skeletal carbonate components, the allochthonous (detrital) micrites is definitively most abundant in comparison to the component directly mineralized (autochthonous micrite) in the cavities of the build-ups. Noteworthy, the percentage of autochthonous micrite, whose early cementation contributes to reinforce the coralline algae framework, is higher in the CBR_2_4_21c build-up than in the CBR_2_3_7c build-up. The average percentages of total cavities, *i.e.* the sum of empty cavities and cavities filled with detrital sediments, are similar in the two build-ups.



### 3.4 Characterization of micromorphology and geochemistry with electron microscopy


SEM observations and EDS microanalyses allowed us to detect the micro/nano-morphologies and the composition of the
main skeletal and non-skeletal components. The presence of pristine micro- and nano-morphology and original
mineralogy of skeletons testify that the carbonate components of the build-ups have not undergone neomorphic processes,
like recrystallization, polymorphic transformation or aggrading neomorphism. In some cases, the skeletal components
display evidence of dissolution process.
Microcavities are filled with sponge spicules and remains of carbonaceous amorphous substances (Fig. 12). Spicules are
mainly oxeas and (sub-) tylostyles, which may be associated with species of the *Cliona* genus, but there are also triactines
and rare tetractines. Spicules are closely intermingled with organic matter and are often well visible cleats with sterrasters
and oxyasters, typical of species of the genus *Erylus* (Fig. 12E and F). Areas close to the corroded boundaries of some
microcavities containing spicules (Fig. 12G-H) also include small carbonate chips (Fig. 12I) seemingly detached from
the encasing skeleton due to mechanical boring activity of the sponges. The substrate of the bioeroded cavities shows the
typical erosion scars (pits) left by the perforating activity of the sponges of the Clionaidae family (Fig. 12D). Spicules
often show circular erosion marks and an enlarged axial canal (Fig. 13A-E) due to silica dissolution caused by high pH
values of the porewater inside the crevices of the coralligenous build-up.

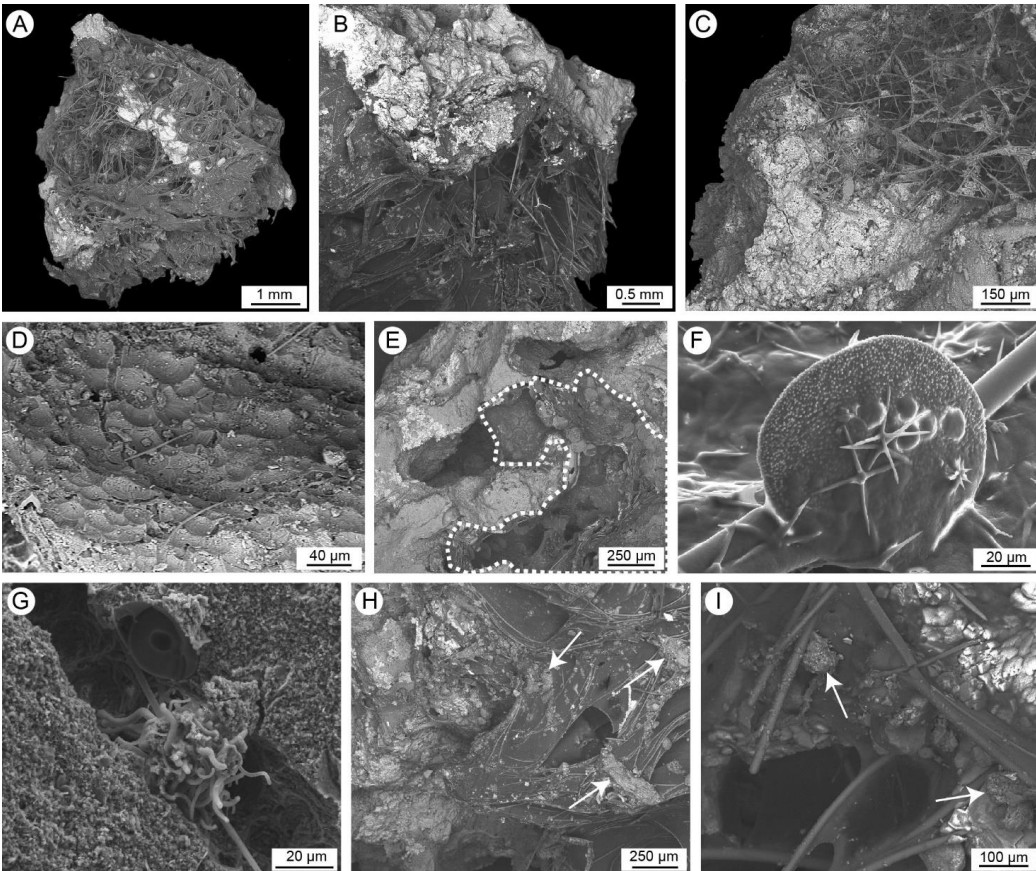


**Figure 12:** SEM images of sponge spicules and remains of carbonaceous amorphous substances. (A-C) Coralligenous
fragments whit pervasive colonization of sponges both on external surface and internal microcavities. (D) Sponge erosion



scars (pits) on a skeletal substrate. (E) Endolithic sponge (dotted line) inside a microcavity of the skeletal framework. (F)
Detail of sterraster and oxyastera of *Erylus* sp. (G) *Cliona vermifera* spiraster associated to a bioeroded cavity. (H)
Spicules and amorphous organic matter in an internal microcavity. (I) Detail of spicules and amorphous organic matter
englobing small carbonate chips (white arrows) detached through sponge bioerosion activity. [B-C: CBR_2_4_21c; A,
D-F, G-I: CBR_2_3_7c].
A high amount of siliceous and rare calcareous sponge spicules (still under study: Bertolino et al. in prep.) has been
recognised in both autochthonous (Fig. 13) and detrital (Fig. 14) micrite pointing to a considerable diversity.
Autochthonous micrite shows micro- to nano-meter anhedral to sub-euhedral crystals engulfed in amorphous organic
material (Fig. 13F) and has a high magnesium calcite (Ca ~91 wt%; Mg ~6.5 wt%) composition with minor terrigenous
components (<2 wt%). Micrite engulfing spicules is well cemented (Fig. 13A, B and E). Peloidal micrite passes to
aphanitic textures when cavities become filled. In comparison to the detrital micrite, the autochthonous micrite engulfing
sponge spicules lacks skeleton fragments or allochthonous components like planktonic foraminifers or coccoliths.

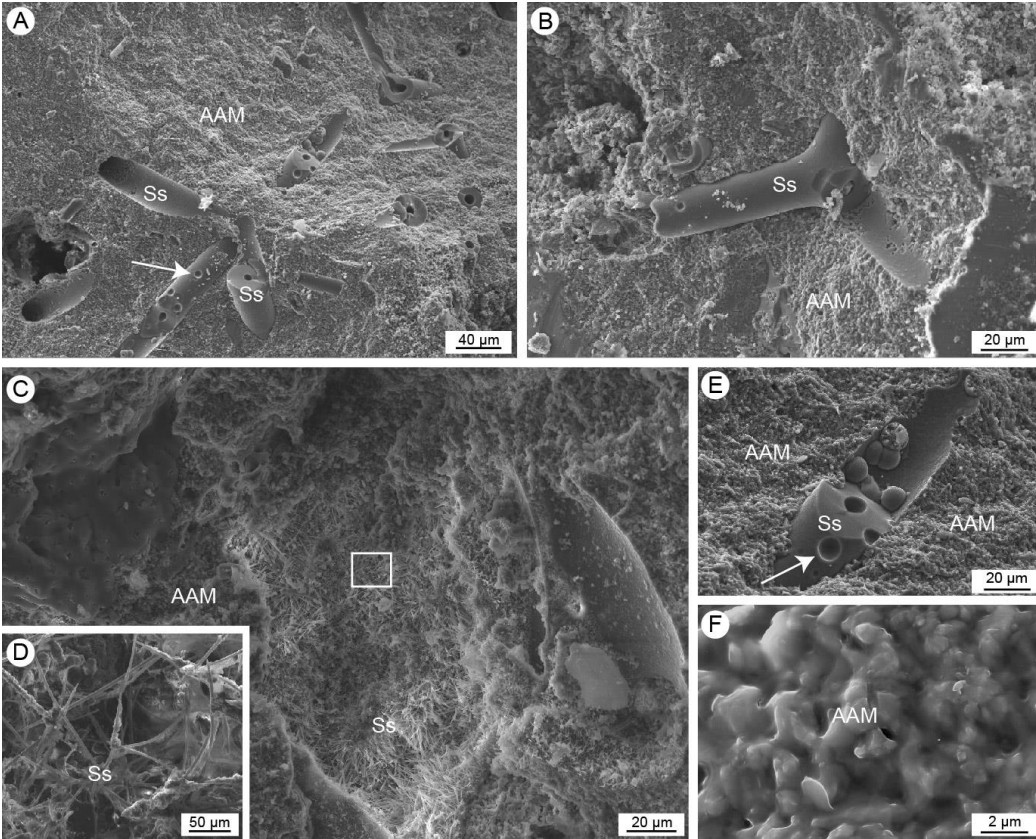

**Figure 13:** (A-F) SEM images of aphanitic autochthonous micrite (AAM) engulfing sponge spicules (Ss). In (A) and (E)
note the well-defined circular boreholes (white arrows) and enlarged axial canal of the spicules. (D) Magnification of the
spicules in (C). (F) Detail of the aphanitic autochthonous micrite showing the micrometer sub-euhedral crystals engulfed
in amorphous organic material. [B-D: CBR_2_4_21c; A, E-F: CBR_2_3_7c].
Detrital micrite shows a heterogeneous composition (Fig. 14), in terms of type and size of the grains, has a magnesium
calcite composition and a high percentage of terrigenous components. It is composed of Ca (~46 wt%), Mg (~3 wt%), Fe
(~6 wt%), K (~3 wt%), a discrete quantity of Si (~26 wt%), Al (~12 wt%) and a lower amount of other elements (each S,
Na, Cl <2 wt%). The presence of sponge spicules in the detrital micrite is constant (Fig. 14), but it incorporates also small




intraclasts and several bioclasts of benthic and planktonic organisms. At the microscale, detrital micrites denote a high
amount of nannoplankton plates (*i.e. Emiliania huxleyi*) (Fig. 14E).

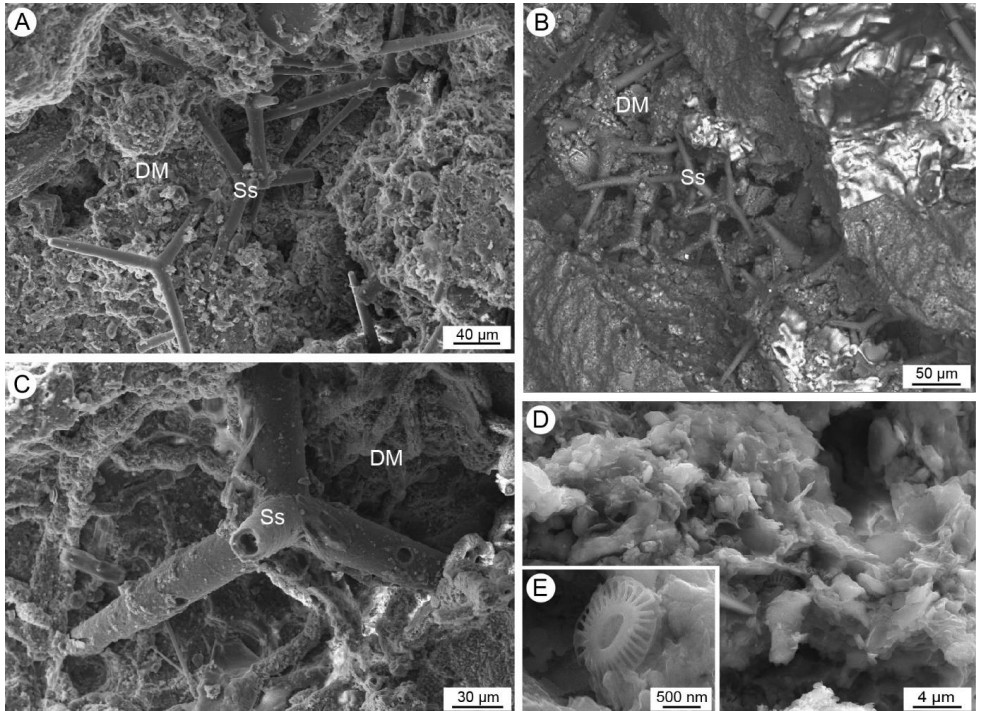


**Figure 14:** (A-C) SEM photos of detrital micrite (DM) engulfing fine intraclasts, bioclasts and sponge spicules (Ss). (D)
Details of the detrital micrite and (E) a nannoplankton plate of *Emiliana huxley*. [B: CBR_2_4_21c; A, C-E:
CBR_2_3_7c].

### 4. Discussion

The analysed coralligenous build-ups from the Ionian Sea are mainly constituted of skeletonized organisms with dominant
coralline algae and subordinate bryozoans and serpulids. CCA form a porous self-sustaining framework whose
stabilization is reinforced by bryozoans and serpulids. The role of foraminifers and corals is negligible. The morphological
growth is influenced by sponges. These organisms are highly diversified and play a triple role: locally affect the direction
of encrustations of the skeletonised builders, weaken the framework through bioerosion processes and induce
precipitation of autochthonous sediments (autochthonous micrite). An important role is played also by the micrite which
could be subdivided into two types: autochthonous, directly produced in the build-ups by organic-induced
biomineralization processes, and allochthonous (or detrital), derived by the accumulation of loose fine particles in the
cavities of the build-ups.

### 4.1 Skeletal builders and framework density

Quantitative counting of the microfacies components demonstrates the role of algae, serpulids and bryozoans as builders
of the two build-ups (see Table 1 and Table 2). These data agree with the indirect quantification obtained through image
analysis and computerized axial tomography by Bracchi et al. (2022) on the same build-ups analysed in this paper. Bracchi
et al. (2022) correlated four density classes with the framework cementation degree and distinguished the different



components, identifying the CCA as dominant. This because computerized axial tomography does not provide a direct association between skeletal and non-skeletal components and the variation of density. In contrast, and despite referring to only one surface for each build-up, the microfacies characterization at microscale confirmed that the density is not directly correlated to specific components but it is linked to the degree of packing of the skeletons (mainly CCA) and to the degree of cementation of non-skeletal carbonate components. Furthermore, it is worth to note that the presence of autochthonous micrite, which cement syndepositionally, contributes to increase the build-ups density, regardless of the nature of the components to which it is associated.

**4.2 Role of the sponges in the coralligenous growth**

Coralligenous growth is the results of the interplay between the building activity and the physical and biological demolition (erosion and dissolution) processes (Garrabou and Ballesteros, 2000; Bressan et al., 2001; Cerrano et al., 2001; Ingrosso et al., 2018; Turicchia et al., 2022). Like for bioconstructions in the Mediterranean and worldwide (Rosell and Uriz, 2002; Evcen and Çınar, 2015; Glynn and Manzello, 2015; Achlatis et al., 2017; Turicchia et al., 2022), boring sponges represent the main cause of bioerosion for the coralligenous build-ups of Marzamemi. The colonization, amount and bioerosion processes of sponges is influenced by temperature, nutrients, turbidity, depth, light, and pH (Schönberg, 2008; Calcinai et al., 2011; Nava and Carballo, 2013; Marlow et al., 2018). Bioerosion has a direct influence on the coralligenous build-ups due to the erosion of the substrate and the skeletal framework which reduces the mechanical stability of the build-ups (Scott et al., 1988), but at the same time it creates new space and shelter for other organisms (Cerrano et al., 2001; Calcinai et al., 2015).

The main boring taxa belong to the family Clionaidae, and especially *Cliona celata, C. schmidtii* and *C. viridis*. Boring and insinuating, cavity-dwelling endolithic sponges, cover considerable proportion of the total biomass, even higher than that of the epibenthic layer, both in coralligenous build-ups (Calcinai et al., 2015) and marine caves (Corriero et al., 2000). The contribution of cryptic fauna should be taken into consideration on Coralligenous studies, considering that the number of sponges occurring outside these build-ups is lower than the number of taxa identified inside them (Calcinai et al., 2015).

The high amount of sponges and their expected diversity fits well with results by Bertolino et al. (2013) revealing the occurrence of 53 insinuating and 10 boring species inside the coralligenous build-ups. Among them, not-perforating encrusting or massive species occupy cavities of the bioconstructions previously formed by boring sponges (Bertolino et al., 2013).

When sponges die, their spicules remain trapped in the cavities of the coralligenous framework, offering the opportunity to recognize the spongofauna over a very long time (Bertolino et al., 2014; 2017a; 2017b; 2019). Spicules of non-eroding sponges from the build-up surfaces may be mixed with spicules of boring and insinuating species in the cavities of the build-ups (Calcinai et al., 2019). In the studied coralligenous build-ups, the pervasive colonization of sponges is testified by the high amount of spicules occurring on the surface and within cavities. Oxeas and tylostyles sponge spicules were observed mixed or grouped by type. It is not clear if the oxeas and other spicules derive to insinuating sponges or if they derive from species (*e.g.* haploslcerids), which encrust the external hard surfaces of the coralligenous. Spicules associated with amorphous organic matter (spongin remains) in empty cavities testify the presence of recently dead endolithic sponges, possibly even after the collection of the samples. Among these, boring sponges are recorded by spicules associated to small chips detached from the substrate in cavities showing irregular edges and corroded surfaces. Most of the spicules are mixed with fine detrital sediment and other bioclasts filling intra- and inter-skeletal holes. These spicules may originate from sponges thriving on the build-up surfaces or their internal niches, and together with those preserved

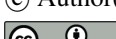



in the autochthonous micrite, they represent the record of past Coralligenous sponges. The characterization and dating of
these components may reveal the ecological evolution and functional role of the sponge associations during the growth
of the build-ups. Sponges seem also to influence the morphological development of the build-ups. They can be often
bioimmurated by encrusting organisms (mainly crustose algae) thus affecting the local growth direction of the carbonate
crusts, that follow the morphologies of the sponges .
**4.3 Origin and role of the micrite sediments in the coralligenous framework**
The micromorphological observations allowed to recognize and investigate the role of the sediment in the Coralligenous.
Of the two main types of sediment detected, i.e. autochthonous and allochthonous (detrital) micrite, the former is produced
*in situ* through biomineralization processes. Biomineralization involves organisms, processes and products and includes
controlled, induced and influenced mineral precipitation (Lowenstam & Weiner, 1989; Riding, 2000, 2011; Benzerara et
al., 2011; Phillips et al., 2013; Anbu et al., 2016; Riding & Virgone, 2020). These different pathways of biomineral
precipitation depend also on the chemical and physical conditions of the environment (Riding, 2011; Riding & Liang,
2005; Deias et al., 2023). The recognition of biominerals, especially those not biologically controlled (like skeletons) but
precipitated via organic mediation in equilibrium with the water medium, can be considered as a remarkable archive,
documenting the presence of non-fossilizable associations as well as their relations with the environmental conditions. In
the Coralligenous, the presence of autochthonous micrite whose precipitation could be linked to microbial metabolic
activity or decaying organic matter mediation, documents communities with low preservation potential but heavily
affecting the development of the build-ups.
The autochthonous micrites detected in the studied coralligenous build-ups show two fabrics: 1) structureless (aphanitic)
and 2) peloidal to clotted peloidal. Both types consist of Mg–calcite and show an intense fluorescence when excited with
UV-light, suggesting a high content in organic matter. Despite these similarities, the two fabrics may have different
origins. The massive presence of spicules engulfed in the aphanitic micrite could indicate a carbonate precipitation in
association with decaying organic substrates of sponges as repeatedly suggested in literature (Leinfelder and Keupp, 1995;
Reitner and Neuweiler, 1995; Reitner et al., 1995; Pickard, 1996; Pratt., 2000; Neuweiler et al., 2003; Reolid, 2007, 2010).
This organic mineralization (organomineralization) is supposed to form via $Ca^{2+}$-binding ability of humic and fulvic
amino acids, derived from organic matter degraded metazoan during early diagenesis (Braga et al., 1995; Neuweiler et
al., 1999, 2007; Wood, 2001; Dupraz et al., 2009). This micrite type is usually darker than the allochthonous micrite, due
to the organic matter content (Warnkle, 1995; Delecat et al., 2001; Delecat and Reitner, 2005). Shen and Neuweiler (2018)
suggested an organomineralization (produced by induced or supported processes), rather than microbial mediation for the
deposition of the autochthonous micrite in the  Ordovician carbonate mounds, formed mainly of calathid–demosponge
(north-west China). A similar process was proposed for the autochthonous micrite mineralized in the biotic crust of
submarine caves of Lesvos (Guido et al., 2019a, 2019b).
The peloidal and clotted peloidal micrite have commonly been linked to anaerobic bacteria activity and represent the main
component of the microfacies recognised in modern and ancient microbialites (Monty, 1976; Chafetz, 1986; Kennard and
James, 1986; Buczynski and Chafetz, 1991; Reitner 1993; Kazmierczak et al., 1996; Dupraz and Strasser, 1999; Folk and
Chafetz, 2000; Riding, 2002; Riding and Tomás 2006; Dupraz et al., 2009; Guido et al., 2013, 2016; Riding et al., 2014).
The microrganisms and metabolic pathways responsible for the formation of clotted and peloidal micrites still remain
unknown in most instances. In the studied build-ups, the scarce peloidal fabric is mainly confined to framework
microcavities, particularly the interior of serpulid tubes and spaces between skeletons. In these suboxic/anoxic confined
microenvironments can flourish anaerobic heterotrophic bacterial communities, as observed in several submarine caves



by Guido et al. (2017a, 2017b, 2019a, 2022). In the Coralligenous, the clotted peloidal micrite is very subordinate in
comparison to aphanitic micrite, and generally it is not associated to sponge spicules, but includes terebellid polychaetes
often intermingled with the autochthonous peloidal micrite. The presence of terebellids associated to peloidal micrite
suggests a close association between these polychaetes and microbial communities. A symbiotic relationship between
terebellids and sulphate-reducing bacteria has been described in confined environments of submarine caves (Guido et al.,
2014, 2022). These authors proposed that in pendant bioconstructions terebellids use the peloids produced by microbial
activity to form their skeletons and the bacteria flourish on the biomass produced by the terebellids and other metazoans.
The occurrence of a similar association in the coralligenous build-ups seems to suggest that this symbiosis is not habitat-
specific, but develops in different marine settings, from open to confined habitats, where conditions of cryptic micro-
environments in the framework of the bioconstructions may promote the development of carbonatogenic microbial
communities.
An early lithification of autochthonous micrite has been suggested by many authors (*e.g.*, Grotzinger and Knoll, 1999;
Reid et al., 2000) as an explanation for the modes of accretion and textures of various types of bioconstructions. The
accretion of the Sicilian Coralligenous was clearly produced by the superposition of different generations of skeletonized
organisms but early lithification of autochthonous micrite, inside primary and secondary cavities, further contributes to
cementation and stabilization of the skeletal framework.
The detrital micrite generally derives from degradation and transport of organism's skeletons, transported from
neighbouring areas, or from erosion of pre-existing bioclastic rocks (Stockman et al., 1967; Tucker and Wright 1990).
The two different types of detrital micrite observed in the studied Coralligenous (i.e. organic or inorganic) differ by the
occurrence or absence of organic material trapped in the muddy sediment, the different degree of fluorescence under UV-
light and the bioclast content. The common lose state of these sediments in the studied build-ups seems to point that their
lithification rate is generally lower in comparison to the autochthonous micrite, and may take place sometime after the
primary framework is formed. The different amount of organic matter, however, may have a role in the lithification
processes. Further knowledge is needed to help the comprehension of the diagenetic dynamic of the detrital micrite
helping to clarify the general growth and morphological development of the coralligenous build-ups.

**4.4 Coralligenous build-ups *vs* Biostalactites: comparison between recent bioconstructions of different marine setting**

Knowledge of the compositional and morphological characterization of the coralligenous build-ups forming along the
open marine sectors of the Mediterranean Sea shelf, allows their comparison with bioconstructions forming in confined
marine settings, such as blind submarine dark and semi-dark caves. These confined environments recently have been
utilized as natural laboratories to study the role of metazoan and microbial communities in forming not-usual
bioconstructions (Guido et al., 2013; Gischler et al., 2017a). Due to the peculiar conditions of cave environments, notably
low water circulation, reduced or null light intensity, oxygen depletion and remarkable oligotrophy, caves are colonized
mainly by cryptic organisms like serpulids and bryozoans, sponges and corals (Harmelin, 1985; Fichez, 1990, 1991).
These organisms may be involved in the formation of small biogenic crusts or larger bioconstructions named biostalactites
that develop under suitable conditions (Onorato et al., 2003; Belmonte et al., 2009, 2020; Guido et al., 2013, 2017b,
2019a, 2022; Sanfilippo et al., 2015; Gischler et al., 2017a, 2017b; Onorato and Belmonte, 2017; Kazanidis et al., 2022).
Serpulids and bryozoans are the main skeletal builders of the biostalactites that are further stabilized by the early
cementation induced by the precipitation of autochthonous peloidal and aphanitic micrites mediated by microbial activity
(Guido et al., 2013; Gischler et al., 2017a, 2017b). Despite the difference in size (from some cm up to 1-2 m), biostalactites




forming in submarine caves of Sicily, Cyprus and Apulia show a uniform style of growth (Guido et al., 2013, 2017b,
2022). In contrast, the biotic crusts forming in the caves from Lesvos island of the Aegean Sea (Fara and Agios Vasilios
caves), show a rich sponge association, widely present both on the surface and in the framework microcavities. There,
the pervasive presence of sponges in almost all the micro-niches of the bioconstructions, play a limiting role in the
development of heterotrophic bacteria (like sulfate reducing bacteria) involved in carbonate precipitation (Guido et al.,
2013, 2019a, 2019b, 2022). The competition for space between sponges and carbonatogenic bacteria has been used to
explain the morphological differences between large biostalactites and biogenic crusts common in the Mediterranean
caves (Guido et al., 2019a, 2019b). In the biotic crusts from Lesvos, sponge spicules are mainly concentrated in detrital
micrite that fills primary cavities. Only sporadically, they are associated with autochthonous micrite, suggesting that
organomineralization linked to soft sponge tissue decay is a relatively minor process in the in-situ production of micrite
(Guido et al., 2019a).
A very similar competition could be suggested for the coralligenous build-ups. Despite the different environmental
conditions, size and morphologies characterizing the centimetre sized biotic crust of Lesvos caves *versus* the some
decimetres to meter-sized coralligenous build-ups of Marzameni, abundant sponges cover pervasively the surfaces and
the internal cavities of both types of bioconstructions. The Coralligenous framework is produced by encrusting red algae
but the primary inter-skeletal porosity derived from the superposition of different generation of skeletons is enhanced by
the bioerosive activity of endolithic organisms. Cryptic cavities could be a site for the settlement of anaerobic bacterial
communities (Guido et al., 2013), but they are occupied by insinuating or perforating endolithic sponges that reduce the
availability of micro-niches favourable for the development of sulfate reducing bacteria, hampering the precipitation of
autochthonous micrite through their metabolic activity.
The decaying of the soft sponge tissue produces a huge amount of spicules that are trapped in cavities together with
detrital fine material. Occasionally, the spicules are engulfed in high organic aphanitic micrite that do not enclose detrital
fragments. This material presumably results from induced and/or supported organomineralization of the soft tissue, like
observed in the biotic crusts of Lesvos caves (Guido et al., 2019a). The same process has been suggested also for the
Ordovician calathid-demosponge carbonate mounds of north-west China (Shen and Neuweiler, 2018).
**5. Conclusions**
The study of the build-ups from the Ionian Sea (Marzamemi area) offered the opportunity to investigate the relationship
between skeletal builders and associated sediments, controlling the general morphology and internal framework of the
coralligenous and allowing the development of a unique ecosystem where peculiar geobiological processes occur, and
make these build-ups natural laboratories useful for the palaeoecological reconstruction of the fossil record.
The studied build-ups are formed mainly of crustose coralline algae, which constitute a self-sustaining skeletal framework
further stabilized by bryozoans and serpulids. The superposition of different generations of builders form a high porous
structure. The porosity is further enhanced by bio-erosive activity of perforating organisms. These discontinuities in the
skeletal framework represent ideal niches for the colonization of cryptic organisms like sponges, bryozoans, serpulids
and microbial communities. Among these, sponges are especially common within internal cavities of the build-ups both
with insinuating and perforating taxa. After decaying of soft tissue, the spicules of these organisms accumulate in the
cavities together with allochthonous micrite and other bioclasts.
Muddy to silty sediments represent the main non-skeletal carbonate component. Sediments consist mainly of
allochthonous (detrital) components derived from external sources or from (bio)erosive processes of the build-up
components. The autochthonous micrite, mineralized directly inside the build-up through organomineralization processes,



represents a minor component. It shows mainly structureless textures and it is associated to sponge spicules. The microbial
derived micrite, showing peloidal to clotted peloidal texture, is rare and fills small intra- or inter-skeletal microcavities.
Actually, sponges colonize the cryptic micro-niches which are ideal microenvironments for the growth of carbonatogenic
bacteria, and the small quantities of autochthonous micrite engulfing the spicules probably results from induced- and
supported- organomineralization of the soft tissue of sponges, rather than from microbial mediation. Despite the
subordinate abundance in comparison to the skeletonized organisms, the occurrence of autochthonous micrite suggests a
possible contribution of this component in cementing and strengthening the skeletal framework due to the early
cementation of this type of micrite.
The formation of microbialites seems to be prevented by the competition between sponges and microbial communities
colonizing the same cryptic spaces. The similar competition among these organisms in the biotic crusts of confined
submarine caves suggests that this relationship is not habitat specific. It may develop in similar microhabitats of different
open to cryptic environments, and could be used for palaeoecological reconstructions and for interpreting the role of
metazoans and microbialite in the fossil build-ups.
**Data availability.** All raw data can be provided by the corresponding authors upon request.
**Author contributions.** Adriano Guido and Mara Cipriani conducted the study and prepared the first draft. All authors
contributed with ideas and in reviewing the manuscript
**Competing interests.** The authors declare that they have no conflict of interest.
**Acknowledgements.** We thank Nunzio Pietralito and SUTTAKKUA diving school (Pachino, SR) and Riccardo Leonardi
(University of Catania) for sampling, and Alessandra Savini, Luca Fallati and Andrea Giulia Varzi (University of Milan-
Bicocca) for providing underwater images. The authors greatly thank Mariano Davoli and Chiara Benedetta Cannata of
the "Microscopy and Microanalysis CM2 Laboratory Center", University of Calabria (Infrastructure SILA), for SEM
analysis. This is the Catania Palaeoecological Research Group contribution n. XXX.
**Financial support.** This work was funded by the Italian Ministry of Research and University – Fondo Integrativo Speciale
per la Ricerca (FISR). project FISR2019_04543 "CRESCIBLUREEF - Grown in the blue: new technologies for
knowledge and conservation of Mediterranean reefs".

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
