# Peer review of "Origin and role of non-skeletal carbonate in coralligenous build-ups: new geobiological perspectives in the biomineralization processes"

_Biogeosciences, 2023_

## Author Response (AR1)

UNIVERSITÀ DELLA CALABRIA

Dipartimento di BIOLOGIA, ECOLOGIA e SCIENZE DELLA TERRA (DiBEST)

Rende 03-11-2023

**Ref.: Ms. No. bg-2023-115**

*"Origin and role of non-skeletal carbonate in coralligenous build-ups: new geobiological perspectives in the biomineralization processes"*

**Biogeosciences**

**Final Associate Editor decision:**

Public justification:

The work was reviewed by two experts. They both agreed that the manuscript was interesting, novel, and scientifically sound. I concur with their assessment. Reviewer #2 provided some very interesting suggestions, which I recommend the authors to consider as they revise their work. Me too, I have some minor comments, as specified below. Overall, the manuscript is appropriate for publication in BG and I would be happy to consider it after minor revisions.

We are grateful for the review of our manuscript, and appreciate the efforts the Editor and Reviewers have made in providing feedback of great value for the revision. The paper has been revised taking into consideration all their suggestions, corrections and comments. answers to the comments and the changes in the text are highlighted in red.

Minor comments:

Missing part of Fig. 1 caption. Please, check and revise.

The Figure caption of Fig. 1 has been modified and revised.

Line 105. Fig. 2C-H cited before 2A-B.

The order of citation of the figures has been revised.

Lines 145-155. In my opinion, the description of primary vs secondary cavities belongs to the Discussion section of the manuscript, where the authors should also include their interpretation of the role, meaning of these cavities. The Results section should be limited to the description of their analyses and observations.

The interpretation of the cavities origin has been moved in the discussion.

Caption Fig. 9 – specie names should be in italics.

The acronyms EDS has been clarified in the methods.

Line 341 – define EDS. Please, add details about EDS analysis in the Methods section of the manuscript.

The acronyms EDS has been clarified in the methods.

Section 3.4 – how did you get such specific element weight %? By EDS? In case, the standardization approach prior to analysis should be clearly described.

The specific weight percentage for each element has been obtained using the EDS. This information has been added in the methods.

General comment for figure captions. In those figures with multiple panels, when the specimens is specified (usually at the end of the caption) always start by listing from A first. E.g., Fig. 8 - [C-D : CBR_2_4_21c; A-B, E- F: CBR_2_3_7c] should be [A-B, E- F: CBR_2_3_7c; C-D: CBR_2_4_21c]

The figure captions of multiple figures have been modified following the suggestion.

**Reviewer#1 comment:**

This is a good paper. The story is interesting, the topic new and faced with an adequate methodology. Although a lot of studies were conducted about the coralligenous bioherms, very few data were until now available about the sediment filling the coralligenous cavities. So, in my opinion this study is particularly welcome.

The group of authors covers different expertise both from a geological and biological point of view. I am particularly happy about the focus on the role of sponges in shaping the coralligenous structure. At this subject, only a comment/suggestion. (i) Sponge spicules are an important component of the micritic sediment of the coralligenous, (ii) substrate chips produced by boring sponges are also a component of the fine sediments, (iii) sponges are (probably with bivalves) the moist important bioerosive element of the coralligenous. All these points were addressed in the manuscript. Nevertheless, from a biodiversity point of view, boring sponges are only few species. Several species are insinuating and have an important structural role in maintain attached fragment of conglomerate detached by the erosive activity. So, in my opinion, different groups of

sponges have antagonistic effects on the shaping coralligenous bioherms. While the bioerosion was studied in detail, very few data are available about the aggregation ability. I encourage the authors to study this aspect in this or in a future research.

Answer:

We are grateful to Referee#1 for the very positive comments on our paper.

The role of sponges in shaping coralligenous build-ups and other recent Mediterranean bioconstructions is new and the constructive *vs* destructive activity of these organisms has been not well investigated yet. In agreement with the Referee's comment, we are already working on the taxonomic characterization of the sponge communities, trying to discriminate boring from insinuating species and quantitatively evaluate their bioconstructive *vs* bioerosive effects on the build-ups. The aggregation ability of the insinuating sponges is a new aspect and we appreciated the suggestion of the Referee. This aspect will be surely added in the new paper we are planning to elaborate on the specific role of sponges on coralligenous bioconstructions.

**Reviewer#2 comment:**

This paper deals with the modern state of bioconstructions of the Mediterranean Sea known as "le coralligène" (I tend to prefer to keep the original french term). This paper is of broad interest, provides new insights, is well structured and pretty well written. Here are some remarks the authors might find useful.

While mentioning a skeletal organism (or a group of organisms) for the first time, please add the basic mineralogy of its skeleton (HMC, LMC, aragonite, especially for the Peyssonneliales, opaline). This appears important (for me) in terms of "reactivity", ageing, preservation and pathways of early diagenesis.

We appreciated the suggestion of the reviewer, and we included the indication about the composition of the main taxa. Our observations also allowed assessing good preservation state for the skeletal components in terms of the early neomorphic processes. We added a short comment about this aspect in the Discussion.

Is there any idea to put a number attached to "high biodiversity", how high is very high and how it is assessed or calculated?

The biodiversity associated with coralligenous has already been evaluated so far. A list of the main taxa related to coralligenous is included in Ballesteros (2006). In the studied site,

Bracchi et al. (2022) evaluated the associated biodiversity with a 2D and 3D approach and we added these information accordingly, as reported from lines 101 to 108.

You might eventually better define the basic attributes of "le coralligène", its distinguishing characters (as a specific system), and its biogeohistoric origin (at least early Cretaceous in my view).

The term Coralligenous (Coralligéne), as the reviewer already indicated, is derived from the French literature (Marion, 1883; Pérès and Picard, 1964), and generally indicates mesophotic bioconstructions of the Mediterranean Sea, primarily built by crustose coralline algae. In recent years, the definition of Coralligenous is under review, also due to the exploration efforts in the deepest part of the shelf. We summarized the main characters of Coralligenous in the first sentences of the Introduction.

The crustose coralline algae have an excellent fossil record from the Early Cretaceous onwards (Aguirre et al., 2010), but build-ups similar to coralligenous are younger dating back at least to the Quaternary. Present-day Coralligenous is Holocene in age (Sartoretto et al., 1996; Bertolino et al., 2017; Basso et al., 2022).

Be very careful with the term "biomineralization" (line 68-70). The work of Trichet and Défarge (1995) on organomineralization might be of value here.

We thank the reviewer for this remark. We refer here to autochthonous micrite as a product of induced or influenced biomineralization, which, for the influenced one, corresponds to the organomineralization processes discussed in Trichet and Défarge (1995). This aspect has been clarified in the text and the suggested references have been added and discussed in the Introduction.

Eventually add more inscriptions and arrows into your figures, you might also try combined figures, one overview and one zoomed in to the necessary detail.

We believe that the figures already contain several inscriptions and arrows that help the reader to understand them. Therefore, we preferred not to add some more, but we made the ones already inserted more visible. Overviews are reported in Figs. 4, 5, 7 and Zoomed-in are reported in Figs. 8, 9, 10, 11.

Eventually replace "micrite" by microcrystalline (in proper place), micrite is sediment (matrix), automicrite is not sediment, it has now history of transport!

We thank the reviewer for this comment, and we know that there is an on-going debate on this topic. In this paper, we used the term micrite *sensu* microcrystalline calcite of Folk (1959) referring to the crystal size of this component. We use the term in a non-genetic descriptive sense adding autochthonous or detrital to underline the difference among the two components. We considered the autochthonous micrite as synonym of automicrite, referring to the component deposited *in situ* through influenced mineralization, whereas the detrital micrite is a "pure" sediment type derived from physical processes. The terminology used in the paper has been clarified in the Introduction and Discussion.

Fluorescence is not a distinguishing attribute of automicrite, microcrystalline sediment and early cements might also fluoresce (Neuweiler et al, 2000, 2003 in GEOLOGY). Instead the locus of fluorophores is crucial. For identifying automicrite a combination of petrographic attributes should be used (see some sort of recent review in Neuweiler at al., 2023 in SEDIMENTOLOGY).....and by now, you should know who I am.

According to the suggestion, the autochthonous micrite was identified through morphology, crystallography and fluorescence. This approach was mandatory because also detrital micrite rich in organic matter (organic detrital micrite = ODM in the paper) shows fluoresce under UV excitation. We thanked the reviewer for the suggesting papers, which we considered in the revision of our manuscript.

Congrats for this paper, again, please be more careful with the terminology
We added specific references to the terminology at which we adhere.

(eventually also more selective and specific with the "geological" references (too much blur there);
We paid attention to the geological terminology as required.

keep it as simple and fundamental as possible (mention the facts, not the interpretations), and admit what we currently do not know) and, in the future, you might dig a bit deeper what concerns the (organo-)chemical attributes and precipitation process of the automicrite in scope. Try to catch it in the making (caught in the act) by using bio-fixation methods. Great material, keep going, very nice, super potential!

We thank the reviewer for the final suggestions and his positive evaluation of the paper. The manuscript describes for the first time new processes and products related to the microcrystalline component inside the coralligenous, about which there are still many things to investigate. One of the approaches, for example, should be the characterization of the organic matter in autochthonous micrite through the biomolecular approach in GC-MS.

Rethink about your title, I suggest…. The constructional architecture of coralligenous build-up.

We appreciate the suggestion, but we prefered to maintain the original title because we believe it matches the content of the paper.

Did not check for typos or the reference list, the editorial office should have a respective software tool.

---

## Author Response (AR2)

**UNIVERSITÀ DELLA CALABRIA**

**Dipartimento di BIOLOGIA, ECOLOGIA e SCIENZE DELLA TERRA (DiBEST)**

Rende 14-11-2023

**Ref.: Ms. No. bg-2023-115**

*"Origin and role of non-skeletal carbonate in coralligenous build-ups: new geobiological perspectives in the biomineralization processes"*

**Biogeosciences**

**Associate editor decision: Publish subject to technical corrections**

The authors addressed all the suggestions received. The manuscript can now be accepted for publication in BG, although its publication is subject to technical revisions. In particular, the authors need to provide additional information about the standard used for EDS analysis and the software used for data collection.

We are grateful to the Associate Editor Chiara Borrelli for the positive evaluation of the revised version of the paper.

The methodology about the SEM/EDS analyses was amended with the required information as follow (lines 146-152):

Selected fragments, used for Scanning Electron Microscopy (SEM) observations and Energy Dispersive X-ray Spectroscopy (EDS) microanalysis, were carbon coated. The SEM apparatus was used is Ultra High Resolution (UHR-SEM) – ZEISS CrossBeam 350. The working condition were: resolution 123 eV, high voltage 10 keV, probe current 100 pA and working distance 11 mm. Mineralogical and chemical compositions were investigated with an EDAX OCTANE Elite Plus - Silicon drift type - $Si_3N_4$ Window apparatus under high voltage 15 keV, probe current 60 mm, working distance 12 mm, take-off angle 40°, and live time 30 sec. The standardless quantitative analysis were checked on SPI #02757-AB serial 4AK standard and were collected through the software AMETEK Apex Suite V2.